# Recent Advances in Peptide Drug Discovery: Novel Strategies and Targeted Protein Degradation

**DOI:** 10.3390/pharmaceutics16111486

**Published:** 2024-11-20

**Authors:** Katarina Vrbnjak, Raj Nayan Sewduth

**Affiliations:** VIB-KU Leuven Center for Cancer Biology (VIB), 3000 Leuven, Belgium

**Keywords:** PROTACs, peptide drug design, peptide drugs, targeted protein degradation, multi-omics, micropeptides, peptide drug delivery

## Abstract

Recent technological advancements, including computer-assisted drug discovery, gene-editing techniques, and high-throughput screening approaches, have greatly expanded the palette of methods for the discovery of peptides available to researchers. These emerging strategies, driven by recent advances in bioinformatics and multi-omics, have significantly improved the efficiency of peptide drug discovery when compared with traditional in vitro and in vivo methods, cutting costs and improving their reliability. An added benefit of peptide-based drugs is the ability to precisely target protein–protein interactions, which are normally a particularly challenging aspect of drug discovery. Another recent breakthrough in this field is targeted protein degradation through proteolysis-targeting chimeras. These revolutionary compounds represent a noteworthy advancement over traditional small-molecule inhibitors due to their unique mechanism of action, which allows for the degradation of specific proteins with unprecedented specificity. The inclusion of a peptide as a protein-of-interest-targeting moiety allows for improved versatility and the possibility of targeting otherwise undruggable proteins. In this review, we discuss various novel wet-lab and computational multi-omic methods for peptide drug discovery, provide an overview of therapeutic agents discovered through these cutting-edge techniques, and discuss the potential for the therapeutic delivery of peptide-based drugs.

## 1. Introduction

As technological progress in science marches forward, the field of drug discovery also needs to undergo a transformative evolution. To successfully develop a small-molecule drug, several early-stage hurdles must be overcome, including target identification and validation, hit identification, and lead molecule optimization. Innovative methods in both theoretical and experimental validation are driving changes in the landscape of early drug discovery tools that are available to researchers. The most notable shift is a switch from the traditional, one-target-at-a-time approach to high-throughput computational methods available today [1]. While traditional methods remain foundational, they often faced limitations in efficiency, cost, and the ability to target complex biological pathways, leading to considerable delays and dramatically increasing costs. It is estimated that bringing a new drug from the bench to the clinic can often incur a cost of up to USD 1 billion, and this process can easily exceed 15 years [2,3]. Furthermore, the failure of a promising drug candidate can occur at any stage of its development, owing to a clinical drug failure rate of around 90% either to issues of toxicity, lack of clinical efficacy, subpar ADME (absorption, distribution, metabolism, excretion) properties, or poor strategic planning [4]. In the case of early drug discovery, established target identification methods frequently fail due to the “undruggability” of target proteins, which are usually molecules without binding pockets that a small molecule may fit into, or multifunctional proteins that cause toxicity upon modulation. During the hit identification phase, screening a large compound library can be prohibitively expensive and time-consuming, and the resulting data may be difficult to analyze. Within the lead optimization process, where promising compounds are refined to improve their drug-like properties, balancing the potency and selectivity of a given compound is often challenging [5].

In response to these bottlenecks, novel methodologies such as computer-assisted drug discovery (CADD), multi-omics analysis, and gene editing techniques have emerged, offering a modern alternative to the drug discovery process. To discover a small drug using computational methods, structure-based and ligand-based approaches can be utilized, applying a large number of techniques that have become much faster and cheaper with the advances in parallel computing [6]. Computational methods have become exponentially more efficient with the advent of artificial intelligence. For instance, machine learning can be used to tackle large volumes of complex biological data and predict the features of a given target or small-molecule drug, harnessing the power of deep neural networks [7]. Network-based methods for drug discovery use known protein interaction networks to identify targets shared between certain diseases, offering solutions to multiple problems simultaneously [8]. A recent therapeutic breakthrough in addressing hard-to-target proteins is targeted protein degradation (TPD). The ubiquitin–proteasome system is leveraged using a proteolysis-targeting chimera (PROTAC) to degrade a specific protein, challenging the long-standing definition of an undruggable protein [9]. This short list represents only a fraction of the novel methods recently developed in order to expand our understanding of early drug discovery and make it both efficient and affordable.

A particularly fast-growing field of drug discovery research is the investigation of peptide-based drugs. These typically 2–50-amino-acid proteins have garnered considerable interest in recent years because of their low toxicity and high selectivity, providing an attractive alternative to traditional small-molecule drugs [10]. Many peptide-based drugs can now be found on the market, most notably insulin and cyclosporine. More recently, tirzepatide, setmelanotide, macimorelin, and many others have been approved by the U.S. Food and Drug Administration [11]. While these small proteins suffer from certain limitations, such as a short plasma half-life due to the presence of native peptidases in the organism, renal clearance, low permeability through biological membranes, and low oral bioavailability [12], there are ongoing efforts to improve their potential as therapeutic agents. Some of these methods include backbone stabilization, side-chain modification, incorporation of non-canonical amino acids, and PEGylation [13]. Peptide drugs are traditionally sourced from canonical short proteins, such as insulin and other hormones, or animal venom, but recent findings have shed light on a large number of non-canonical peptides that are encoded by small ORFs present in molecules such as long non-coding RNA. Although these peptides have been historically overlooked, evidence shows that they are pervasively translated and play important roles in many biological processes [14]. This review will further explore some of the most recent highlights of cutting-edge techniques used in the early drug discovery of peptide-based drugs, showcasing an array of real-world examples of compounds discovered and developed through such strategies. We will, in particular, focus on multi-omic methods that combine various approaches, and will discuss their practical applications in more detail. We will also describe the novel applications of targeted protein degradation as they relate to peptide-based compounds.

## 2. Novel Peptide Drug Discovery Methods and Practical Applications

### 2.1. Wet-Lab Methods

When designing a peptide that will allow for the targeting of a given protein, two strategies may be used, depending on whether an existing ligand for this protein is known or not. For example, if a biologically occurring peptide has a known receptor or target, synthetic chemistry may be employed to optimize its properties and improve its therapeutic profile. However, if a peptide ligand for a given protein of interest is unknown, de novo peptide ligand discovery must be performed [15]. Historically, peptide drug discovery has been performed via screening libraries, using methods such as phage, bacterial, yeast, and mammalian cell surface displays, as well as acellular displays such as mRNA and cDNA displays. These methods use peptide libraries expressed on either coat proteins of a phage, the outer membrane of a bacterium, or the cell wall of yeast to identify binding proteins. mRNA display links a protein to its parent mRNA molecule in vitro, while cDNA display builds on this concept by converting the mRNA in the approach to cDNA [16]. Peptides may be identified from natural products, either as bioactive peptides from sources such as animal venom, or as non-ribosomal peptides produced by peptide synthetases that may be found in some bacteria and fungi [17]. This chapter will describe some of the most recent highlights in peptide drug discovery that primarily use wet-lab-based methods.

Tumor-targeting peptides (TTPs), alongside cell-penetrating peptides (CPPs), are an attractive topic in targeted therapy for cancers. One example of their utilization is in the design of peptide-drug conjugates to precisely deliver therapeutic payloads. This was recently demonstrated in a study that conjugated paclitaxel with a TPP and CPP, leading to diminished paclitaxel resistance, decreased normal cell cytotoxicity in vitro, and greater antitumor efficacy for breast cancer in vivo [18]. Cell-penetrating peptides have also inspired the development of antifungal agents as demonstrated in another study. Octaarginine, a classical CPP, was elongated and modified with glutamic acid residues, resulting in a stable peptide polymer that exhibits potent and accurate antifungal activity [19]. Recently, a KRAS inhibitory clinical compound called LUNA18 was identified through the use of a novel platform technology that utilizes cyclic peptides. The researchers used mRNA display libraries because of the extremely large number of unique peptides that may be generated through such an approach. The peptides were then cyclized via the N-terminal amine and carboxylic acid group in the aspartic acid side chain. One of the hits acquired through this method, after lead optimization, was then shown to inhibit KRAS in vitro and in vivo, inhibiting cancer cell growth across a variety of cell lines [20]. A novel method for disulfide-rich peptide drug discovery focuses on multicyclic peptides, a remarkably stable class of small proteins constrained by disulfide bonds. This method relies on the CPXXC motif as a disulfide-directing motif to be harnessed in the design of multicyclic peptide scaffolds, which allows for the development of novel libraries of such bioactive peptides [21]. In mass spectrometry-based techniques, affinity selection has been used to discover peptide binders using a synthetic peptide library, offering a promising approach to expedite peptide drug discovery. Researchers developed a platform that combines bio-layer interferometry with high-resolution nanoscale liquid chromatography-tandem mass spectrometry, demonstrating that this approach exhibited high selectivity to binder proteins with high specificity [22]. As a strategy to explore the possibilities of bioactivity of a given peptide, stereorandomization has been proposed, and in one study, solid-phase peptide synthesis was used to generate numerous stereorandomized peptides from known antimicrobial peptides. These modified molecules, in some instances, exhibit a distinct and improved therapeutic profile to the original, non-stereorandomized versions. This approach has strong potential to expand the therapeutic range of known peptide drugs. [23]. In an effort to improve the efficacy and drug-like properties of cationic antimicrobial peptides (CAMPs), researchers have developed flavonoid-based and xanthone-based peptidomimetics of known CAMPs. The resulting molecules were shown to retain their antimicrobial properties while overcoming drug resistance [24].

A new strategy for the investigation of macrocyclic peptides has integrated a bacteriophage display library and peptide cyclization. As macrocyclic peptides are an attractive research topic for protein–protein interactions due to their rigidity and potential to interact with proteins without a binding pocket, researchers have developed a platform named MOrPH-PhD to screen a large library of these peptides displayed on M13 phages, followed by noncanonical amino acid-mediated peptide cyclization. This led to the discovery of several high-affinity binders and inhibitors of various proteins, establishing a platform for the generation and functional exploration of macrocyclic peptides [25]. The phage display peptide library was also used to discover a CD24/Siglec-10 blocking peptide, and researchers improved on the peptide’s original design by changing L-amino acids into D-amino acids, which decreased the peptide’s sensitivity to proteases and lowered its propensity for hydrolysis and degradation. This modified peptide was further shown to enhance tumor cell phagocytosis and to inhibit tumor growth when combined with radiotherapy in several different cancer cell lines [26]. A novel method for cyclic peptide libraries has recently been developed, utilizing the one-peptide-on-one-bead technology. This method allows for easy sequencing of cyclic peptides through mass spectrometry because of the common structure of the 90 μm PEG-grafted polystyrene beads used, as well as permitting the identification of interacting proteins due to the relatively high amount of peptide that may be carried on one bead [27].

A plentiful and underutilized source of bioactive peptides, which has only lately come to light thanks to recent advances in omics-based technologies, is the non-coding genome. The parts of the genome historically labeled as junk, such as long non-coding RNA (lncRNA), intronic sequences, and pseudogenes, have been found to contain small open reading frames (sORFs) that are actively translated to form micropeptides. These micropeptides have come under scrutiny recently, and insights into their mechanisms have yielded a wealth of small proteins that play important roles in all parts of the cellular machinery [28]. Particular emphasis has been placed on the research of bioactive peptides with a role in disease, and novel tools have been developed to validate and study these peptides. For instance, the 53-amino-acid (aa) micropeptide HOXB-AS3, encoded by an lncRNA, the application of which was found to suppress cancer cell growth and migration both in vitro and in vivo, was discovered through ribosome footprinting, or Ribo-Seq. This method uses deep sequencing to identify which mRNA segments are actively translated, i.e., bound to ribosomes, and uses three-nucleotide periodicity to filter out sporadic ribosome-binding events [29,30]. Advances in mass spectrometry have also facilitated micropeptide research efforts, with miPEP133, a 133-aa peptide encoded by a pri-miRNA transcript, being discovered through this method. This peptide was found to have tumor-suppressive qualities in ovarian cancer [31]. Bioinformatics tools, further discussed in the next chapter, can also help with the discovery of these elusive peptides. In one study, a lncRNA-encoded peptide named ASRPS was shown to inhibit the angiogenesis of triple-negative breast cancer, and the researchers found and validated this peptide through a combination of ORFfinder and Ribo-Seq data [32].

### 2.2. Computational Methods

CADD and virtual screening in drug discovery are powerful tools, owing to their cost and time efficiency, which often beats traditional drug discovery methods, and their versatility, which has accelerated rational drug design. There are a number of well-established and popular methods for CADD, encompassing both ligand-based and structure-based approaches. Ligand-based drug design methods, such as pharmacophore modeling and quantitative structure–activity relationship (QSAR), are used when an active ligand is already known, and can be beneficial for rational peptide drug optimization. For structure-based drug design, molecular docking can be used to model the interaction between a protein of interest and its ligand peptide, providing information about how this interaction behaves in a 3D space [33,34]. However, recent technological advances have transformed the landscape of available bioinformatic tools, at the forefront of which is artificial intelligence (AI). With its capability to process vast amounts of data, this technology is becoming more frequently used in the field of drug discovery. The many recent advances in predictive and generative modeling, resulting in easily applicable and free tools such as Google’s AlphaFold, have heralded a revolution in the discovery of peptide drugs [35]. Machine learning (ML) has similarly been widely used in recent years, with researchers harnessing the power of deep learning algorithms to tackle issues such as lead discovery and drug repurposing. This chapter will focus on the recent highlights in the bioinformatic side of peptide drug discovery. Large language models, in particular, have been beneficial in this field, and protein language models have been developed to study the function and structure of proteins based only on their amino acid sequences [36,37]. This has led to the utilization of fine-tuned language models to predict protein-peptide interactions based on nothing but amino acid sequences, which promises to advance the rational design of novel therapeutic peptides [38]. Deep learning was used to predict a putative peptide-binding residue in a given protein, resulting in a tool known as PepCNN that combines a convolutional neural network and a protein language model. This tool, given a protein sequence, is able to predict the residues in it that are capable of binding a peptide, streamlining the search for an active site that may be targeted in a therapeutic approach [39]. Thanks to the use of deep temporal convolutional networks and transfer learning techniques, datasets of antifungal and antibacterial peptides were used to train a machine learning model, and researchers found that it predicts peptides bearing antifungal properties with high accuracy [40].

A deep-learning-based model, using two binary classification models and one multi-classification model, was also utilized in the discovery of antimicrobial peptides, and twelve peptides with predicted antimicrobial properties were chosen from a library of 30,000 random peptides. Further testing confirmed that three candidate peptides out of the twelve predicted ones exhibited antimicrobial activity both in vitro and in vivo [41]. A machine learning pipeline was recently developed to screen for antimicrobial peptides, in particular, bacteriocins, the antimicrobial peptides of bacteria. The authors, using a learning set of 343 known bacteriocins, were able to select 16 peptides from their predicted set. They then further selected for putative bacteriocins with satisfying charge, helicity, and hydrophobic moment scores. After functional testing, it was found that several of these peptides inhibited bacterial growth in vitro, while having minimal effect on mammalian cells [42]. Antimicrobial peptides were also investigated using machine learning when researchers developed a prediction method named Deep-AmPEP30. A deep convolutional neural network and reduced amino acid composition were used to build a pipeline that predicts short antimicrobial peptides with high accuracy. When this pipeline was applied, the authors found a top-ranking short peptide that was able to dramatically inhibit growth in several species of bacteria [43]. This highlights the potential that similar tools have in drug research, particularly as antibiotic resistance becomes more common. Machine learning has allowed for the integration of generic peptide prediction and the identification of their physicochemical properties. In one study, a random forest algorithm was used in combination with these two variables, leading to a successful description of eight therapeutic peptides and highlighting the potential of this approach in the classification of further therapeutic peptides [44]. Anticancer peptides were predicted in a recent study via a low-dimensional machine learning model, which aimed to sidestep the challenges that arise from using high-dimensional features in machine learning. The authors used 19 dimensions in their model and predicted that a number of features distinguish anticancer peptides from the rest, namely, polarization, hydrophobicity, secondary structure, and the glycine, leucine, cysteine, and lysine content [45]. Fragment screening, the method that identifies chemical fragments that can bind to a given protein, was used in an experimental peptide-tethering strategy in a recent study. Researchers used a rational design approach to improve the binding of the MLL peptide to the KIX domain of a protein of interest by modifying its side chains to better fit into the cavity within which the peptide normally binds. This method resulted in a 2000-fold improvement in binding capability for the peptide and is an example of how peptidomimetics offer a promising approach to drug development in medicinal chemistry [46]. Artificial intelligence is also helping to improve the well-established method of peptide molecular docking. Researchers previously used AlphaFold-Multimer for peptide-protein interaction prediction, and a comparison with DockQ showed that this approach is successful [47]. The recently released AlphaFold3, with its capability to predict peptide-protein interactions with high accuracy, will surely be instrumental in further molecular docking studies [48]. Meta-learning, the training of AI models to improve their efficacy through training with tasks instead of with samples, has been utilized in bioactive peptide discovery. A recent method used meta-learning to develop an ML model that works remarkably well for the prediction of IL-6-inducing peptides [49].

The discovery of non-canonical micropeptides that have the potential to be used in a therapeutic approach can be challenging because of their low expression levels and prohibitively small size. To this end, many bioinformatic tools for micropeptide prediction have been developed. Some of the recent highlights include RNAsamba, which uses a neural network architecture to predict sORFs and recognizes the Kozak consensus sequence necessary for the translation process [50]. MiPepid is a machine learning tool trained on a database of known small proteins, which is able to predict sORFs based only on the aa sequence with 96% accuracy [51]. Ribosome profiling data is dependent on bioinformatics, and tools such as RiboCode, which leverage 3-nucleotide periodicity to annotate the translatome, are vital to the deconvolution of Ribo-Seq data [52]. Bioinformatic approaches for non-canonical micropeptide prediction have been successfully implemented in the discovery of therapeutic peptides, as is the case with the peptide known as CIP2A-BP. Researchers identified this lncRNA-encoded micropeptide through a bioinformatic analysis of Ribo-Seq and RNA-seq datasets using the tools cutadapt, TopHat2, Cufflinks, and Cuffdiff. Functional testing further showed that this peptide inhibits the migration and invasion of triple-negative breast cancer cells both in vitro and in vivo [53].

### 2.3. Peptide Based PROTACs

Molecular targets may be considered untargetable or undruggable by traditional means because of several reasons. Structurally, a protein may have a distinct lack of a druggable binding hydrophobic pocket, which makes it difficult to effectively bind a small molecule. This is frequently the case for non-enzymatic proteins. A number of proteins have intrinsically disordered regions, which thwart drug design and are especially true for transcription factors [54]. Protein–protein interactions are notoriously hard to target, owing to the large and flat surfaces produced by such interactions, which frequently lack a classical binding pocket. The localization of a given protein can also be an issue, as intracellular and nuclear targets can be harder to precisely reach with a small-molecule drug, because of the membrane barrier and the cell’s efflux mechanisms. Finally, a large portion of undruggable proteins has an extensive and complex mechanism of action that includes a large number of downstream effectors, and targeting them may therefore result in significant toxicity [55]. Traditional examples of undruggable targets include RAS family members, MYC, and TP53 [56]. Despite these challenges, a new method for tackling these stubborn proteins has recently emerged. Targeted protein degradation has been made possible through PROTAC technology. These small chimeric molecules harness the ubiquitin system of the cell by simultaneously binding the protein of interest and E3-ubiquitin ligase, resulting in precise degradation of the target protein. Upon PROTAC binding, the E3-ubiquitin ligase complex acts on the protein of interest. The protein is then poly-ubiquitinated and recognized by the proteasome, which then digests it [57] (Figure 1). This approach is remarkably selective and can be adapted to target a wide array of proteins, making it one of the most exciting therapeutic approaches discovered recently (Table 1).

It is possible and advantageous to use peptides as the targeting moiety to bind to a protein of interest, primarily since peptide-based PROTACs do not necessitate binding pockets, unlike small molecule-based ones [58]. For example, peptide-based PROTACs have been used as regulators of FOXP3, a hallmark of regulatory T cells that plays an important role in immune tolerance. It is known that its degradation helps with effective anti-tumor immunity, and with this in mind, researchers designed PROTAC molecules based on a 15-aa peptide inhibitor of this protein that was previously discovered by a phage-displayed library. The peptide was bound to the VHL E3 ligase ligand with a linker, and the authors go on to show that the resulting PROTAC can regulate FOXP3 expression in regulatory T cells [59]. In prostate cancer, the protein p300 is known to promote oncogenic signaling pathways and contribute to a more aggressive phenotype. A peptide antagonist sequence specific to the CH1 domain of p300 was bound to an MDM2-targeting peptide sequence, and the resulting PROTAC was shown to effectively degrade p300 and inhibit prostate tumor growth in vitro and in vivo. To develop the peptide sequence, the authors implemented an AI-based approach, using Rosetta’s virtual hot-spot amino acid screening [60]. AI was also used to develop a peptide PROTAC to target the androgen receptor and develop a therapeutic approach for androgenetic alopecia. The authors used ProteinMPNN to design potential binding skeletons for the androgen receptor and VHL, then proceeded to design binding sequences with RFdiffusion. Validation was performed with Alphafold2, and the linker length was determined using ZDOCK. Once implemented, this approach was shown to significantly induce hair follicle cell regeneration in vivo [61]. As a possible therapy for pancreatic cancer, the oncoprotein CREPT was targeted via a peptide-based PROTAC. The targeting peptide for CREPT was rationally designed by investigating this protein’s 3D structure, which led to the prediction that one of its domains contains a motif that will be able to form a homodimer. The motif was then chosen as the targeting arm of the chimeric molecule, while a VHL ligand constituted the other arm. The authors also included a cell-penetrating peptide to improve the permeability of this construct. It was then shown that the PROTAC is able to both permeate into pancreatic cancer cells and degrade its target, leading to a significant inhibition of cancer cell proliferation in vitro [62].

The possible shortcomings of peptide-based PROTACs, such as their poor cell permeability, low stability, and occasionally subpar potency, were circumvented in one study through the use of gold nanoclusters. Researchers developed a peptide-based chimeric molecule to bind HER2 and the E3 ubiquitin ligase component, cereblon. The peptide that binds to HER2 was initially found through a random peptide phage library screening in a previous study. The PROTAC was conjugated to gold nanoclusters through gold–sulfur coordination, and it was shown that this approach resulted in HER2 degradation and cancer cell cytotoxicity both in vitro and in vivo [63]. Stapled peptide-based PROTACs were used to target DHHC3 in cervical cancer, thereby inhibiting the PD-1/PD-L1 pathway, suggesting a therapeutic approach. The DHHC3-binding peptide was chemically stapled using non-natural amino acids to increase its stability, affinity, and confer the possibility of crossing the cell membrane. It was then fused with a linker and E3 ligase binder, and this chimera was shown to degrade DHHC3 in vitro [64]. Stapled peptides were also utilized in a study aiming to degrade MDM2/MDMX, leading to the stabilization of p53 and resulting in antitumor activity. A peptide with potent dual specificity for these two proteins was identified in a previous study through phage display techniques and systematic mutational analysis. The peptide was chemically stapled at a single helix turn, combined with a VHL ligand to form a PROTAC, and its application was then shown to inhibit colorectal cancer cell proliferation both in vitro and in vivo [65]. Another effective strategy for improving peptide stability and cell permeability is their cyclization, a method that was employed to develop a peptide-based PROTAC to target estrogen receptor alpha for breast cancer therapy. The binding peptide, described in a previous study, was cyclized using a cross-linked aspartic acid strategy. This peptide was linked to a VHL ligand using a 6-aminocaproic acid linker. The resulting PROTAC construct was found to significantly induce apoptosis for breast cancer cells in vitro and in vivo [66]. Chemical stapling in peptide-based PROTAC development was also successfully used to design a PROTAC targeting estrogen receptor alpha. The peptide was first rationally designed in a previous study, in which researchers noted that nuclear receptors contain a hydrophobic groove, which can act as a motif to bind a peptide. The resulting peptide, named PERML, was later found to be cleaved at a disulfide bond inside the cell and quickly degraded. In response to this, hydrocarbon stapling was used to stabilize the helix of PERML, thereby significantly improving its stability. The PROTAC was designed with the stapled PERML and an IAP ligand, and the resulting chimeric molecule was then shown to induce estrogen receptor alpha degradation [67]. A novel approach to increase intracellular stability within a peptide-based PROTAC was used when researchers incorporated a beta-hairpin sequence motif in their Tau-targeting PROTAC design. The Tau-binding peptide derived from beta-tubulin was fused to a beta-hairpin sequence and compared to a classical PROTAC design. This approach was found to both effectively degrade Tau in vitro in a proteasome-dependent manner and be more stable over time than the PROTAC that only had a linker, a VHL-recruiting degron, and a cell-penetrating peptide [68]. In acute lymphoblastic leukemia, researchers sought to specifically degrade GPX4, a protein that is highly expressed in cancer and correlates with a poor prognosis. To this end, they used the ubiquitin ligase MDM2, since it is also highly expressed in acute lymphoblastic leukemia and would provide a higher GPX4 degradation rate in cancer cells as opposed to normal tissue. The authors used phage display to discover the GPX4-binding peptide and fine-tuned its structure using Rosetta. For the MDM2-linking part, they used a previously published binding sequence, and the two parts were connected into a chimeric PROTAC molecule. The drug was loaded into gold nanoparticles and was shown to induce GPX4 degradation in vitro, as well as suppress proliferation of cancer cells [69]. Estrogen receptor alpha was targeted in another study by a PROTAC based on a cell-permeable stabilized peptide. The known receptor-targeting peptide was chemically constrained using an N-terminal aspartic acid cross-linking strategy. The peptide was then bound with a 6-aminohexanoic acid linker to a hydroxyproline-containing pentapeptide, which in turn binds the VHL E3 ubiquitin ligase. The researchers then showed that treatment with this PROTAC degrades its target in a proteasome-dependent manner, kills breast cancer cells in vitro, and inhibits their growth in vivo [70]. In prostate cancer, the androgen receptor has been targeted with a peptide-based PROTAC, the structure of which was elucidated via AI-aided peptide drug design. This peptide was linked by a flexible linker sequence to an MDM2-targeting sequence and loaded into gold nanoparticles. The application of the compound was able to degrade the androgen receptor and inhibit tumor growth in vivo [71].

### 2.4. Advances in Peptide-Based Drug Delivery

To address the issues that are commonly associated with peptide therapeutics and that prevent them from reaching the clinic, such as inefficient cell permeability and quick renal clearance, it is paramount to develop appropriate delivery systems. These frequently include a variety of nanocarriers, such as nanoparticles or liposomal nanocarriers, into which peptides are loaded and that can easily bypass the cellular membrane and deliver their payload inside the cell, overcoming several challenges of peptide-based drugs. There are a number of recent publications expanding on this concept. For example, there have been improvements in the technology of solid lipid nanoparticles, a system that is composed of a solid lipid matrix core into which a lipophilic drug is loaded and covered with a surfactant layer to enhance stability. Once administered, this matrix erodes over time and releases the drug from its core [72]. This approach has shown promising results when loading peptide-based drugs for delivery, as was shown with insulin [73]. Polylactic-co-glycolic acid (PLGA) nanoparticles are an attractive and biodegradable delivery method for drugs and are approved by the U.S. Food and Drug Administration for medical applications. Recently, a study has shown that they can efficiently encapsulate peptides, and these nanoparticles were used to develop a seasonal influenza vaccine containing multi-epitope peptides [74]. Another example of PLGA nanoparticles being used to encapsulate peptides is in wound healing, where researchers successfully synthesized such nanoparticles with the tripeptide glycine-L-histidine-L-lysine, known to stimulate healing of injured tissue. The peptide was conjugated with L-carnitine, loaded into PLGA particles, and shown to offer significant skin repair efficiency [75]. Gold nanoparticles can be conjugated with a variety of drugs to facilitate their release inside the body, and, in particular, can be used for peptides to protect them from degradation and improve bioavailability. This approach was done with auto-antigenic peptides, which were used in a mouse model to prevent autoimmune diabetes. Although these peptides are poorly soluble in aqueous media, conjugation with gold nanoparticles was found to enhance delivery of auto-antigenic peptides to lymphoid organs [76]. Chitosan-gellan gum nanoparticles have been successfully used to deliver peptide-based drugs to the colon. These two natural polysaccharides are mucoadhesive and therefore suitable for colon-specific delivery, and researchers have shown that nanoparticles assembled from these components have a favorable uptake and controlled release rate for polymyxin B, an antimicrobial peptide [77].

## 3. Discussion

Peptides as therapeutics have numerous inherent advantages. Their small size makes them inexpensive and convenient to synthesize, and also makes them simple to modify with methods such as cyclization or chemical stapling. These modifications can significantly improve the stability and permeability of peptide-based drugs. Due to their relatively large surface area when compared to small-molecule drugs, peptides have high selectivity for their target protein, and this specificity reduces off-target effects. They have good large shallow surface adsorption, surpassing the traditional need for deep binding pockets in target proteins. Peptides can hold structural motifs like alpha helices, beta sheets, and gamma turns, so they may have structural complexity that is absent in small molecules, leading to a more selective final ligand. They may be easily conjugated with a number of other molecules, leading to combinations such as peptide-drug conjugates or PEGylated peptides to improve plasma half-life. A recent study harnessed the selectivity of enzymes and used a directed evolution strategy to engineer enzymes that can modify peptides in a site-selective manner, which hints at a plethora of strategies to improve the potency of existing bioactive peptides [78]. Peptides also tend to have low immunogenicity and toxicity. This all highlights the unused potential that these small proteins are capable of bringing to the table in the pharmaceutical industry [79,80,81].

Despite these exciting advantages, we cannot be blind to the intrinsic shortcomings of peptide-based drugs. They are known to have low permeability across biological membranes, such as the cell membrane, which in many cases precludes their potential for intracellular targeting. Cellular access of peptide drugs relies on a peptide’s charge and lipophilicity, with high lipophilicity and a positive net charge being hallmarks of the cell penetrative properties of a peptide [82]. This property also means that a large subset of peptides cannot easily cross the gut lining and blood-brain barrier, leading to difficulties in drug delivery for patients and challenges in bioavailability. The inability to cross the gut lining also means that most peptide drugs need to be delivered by injection and cannot be modified to be administered orally, which lowers patient compliance. There are efforts to circumvent this: loading peptide drugs into carriers such as gold or PLGA nanoparticles, or even the creation of cell-penetrating peptide conjugates, can facilitate active transport for peptides across membranes, but research in this area still needs improvement. Peptides tend to have low stability and are susceptible to proteolysis by proteases or peptidases, due to the amide bonds in their structures, leading to their rapid degradation in the organism once administered. This metabolic stability issue can be addressed with a number of methods, such as chemical stapling, cyclization, N-term acetylation or C-term amidation to protect against degradation, replacement of L-aa with D-aa, and the inclusion of non-proteinogenic amino acids in their structure to increase rigidity and lower the availability of peptides modified in this manner to proteolytic enzymes. The problem remains that stapled and otherwise modified peptides may not always work, depending on the structure of the protein of interest and the binding site. Peptides are also known to have very short plasma half-lives and to undergo rapid renal clearance, measured in minutes, since their small size allows them to easily pass through the glomeruli. This can be tackled by the conjugation of a peptide to albumin or another protein with a long circulation time, by lipidation, or by conjugation with large biocompatible polymers such as PEGylation. More challenges exist in this field, such as solubility issues and tissue heterogeneity. Solubility in aqueous media, such as blood, may be low for peptides with a large percentage of hydrophobic amino acid residues, thus limiting their bioavailability as drug compounds and potentially causing peptide aggregation. Peptides often have low toxicity because of their specificity, but that does not mean that unwanted toxicity does not happen in some instances, preventing many peptide drugs from reaching the clinic. The variable cost of the manufacture of peptides is a limiting step to their mass production, since, although synthetic peptides are normally easy and relatively cheap to produce, small molecule drugs are cheaper still [83,84] (Table 2). Artificial intelligence, with its ability to characterize and predict a vast array of peptide-based drugs, is an attractive topic for peptide research and could potentially not only save time and money in the drug development process, but also serve to optimize delivery methods for such compounds. However, current AI models possess biases and limitations that restrict their applicability in this manner. Due to the inherent complexity of biological data, limited existing information on peptide drugs which in turn limits the training data for AI models, and as-of-yet unfixed tendency for AI hallucinations and inaccurate results, generative AI is not yet capable of directly predicting therapeutic outcomes for peptide-based drugs. Other issues include biases in current training data and concerns about the data privacy of large language models, all of which will need to be addressed in order to improve real-world applicability of AI in drug discovery and delivery [85].

While PROTACs represent a fascinating new chapter in TPD, a topic that has given a new hope to the research on undruggable proteins, peptide-based PROTACs combine the novel mechanism of ubiquitin ligase recruitment with the unique contributions of a peptide-based therapeutic. Peptides’ high selectivity and reduced need for a deep binding pocket allow for the targeting of a much larger set of proteins than is traditionally possible with small-molecule drugs. The dual-function nature of the PROTAC then allows for the degradation of the target protein, harnessing the ubiquitin-proteasome pathway and ensuring that the protein of interest is degraded in a specific and direct manner. The versatility of possible peptide conformations and features opens the door to a vast array of protein-targeting moieties—even if the peptide-protein binding itself does not lead to a therapeutic effect, it is then possible to construct a peptide-based degrader, a strategy which has the potential to rewrite the definition of an undruggable protein. In this approach, the peptide does not need to bind to a biologically active site on the target protein, expanding the range of accessible targets. Traditional PROTACs require the target protein to contain a small-molecule binding surface, a requirement that is overcome by peptide-based degraders due to the ability of peptides to bind to a diverse group of targets even when a binding pocket is not available. The construction and application of peptide-based PROTACs come with certain challenges, both those related to the peptide part and those inherent to PROTACs. We have already discussed the shortcomings of peptides, such as their low cellular permeability, propensity for degradation, and low stability. When coupled with an E3-targeting moiety, the resulting molecule has a high molecular weight, which in many cases limits cellular permeability and causes poor pharmacokinetic properties. These molecules also tend to have a large polar surface area, which similarly interferes with cell membrane permeability and causes reduced absorption and lower bioavailability. Their aqueous solubility tends to be low, posing a further challenge [86,87]. However, when PROTAC molecules are loaded into carriers such as liposomes or nanoparticles, their intracellular delivery is drastically improved, hinting at a promising strategy for clinical use.

Drug delivery of peptides is an evolving topic and one of the most important hurdles to their therapeutic use. Due to the challenges that peptide drugs face regarding permeability, stability, bioavailability, and plasma half-life, drug delivery systems are actively being developed to facilitate the use of peptide-based drugs in the clinic [88]. Peptide drugs may be loaded into nanocarriers, such as liposomes, micelles, and polymeric or inorganic nanoparticles such as gold or silica, forming conjugates that can overcome the limitations of these small proteins. These small carriers have a large surface area and normally do not exceed 100 nm in size. Loading peptide drugs into nanoparticle-based systems offers the advantage of improved stability and solubility of the encapsulated molecule, prolonged blood circulation time, and easier transport through biological membranes, improving the pharmacological profile of the cargo and making this an attractive approach. Considerable challenges still exist with this strategy, mostly centered on the stability of the particles themselves. Additionally, some nanoparticles, especially those made of non-biodegradable materials such as iron oxide, can accumulate in tissues and cause toxicity or immune responses [89,90]. Cell-penetrating peptide conjugation is another popular method for improving peptide permeability. CPPs can easily translocate through a cell membrane and deliver a molecule that may normally be blocked by the selective impermeability of the membrane, which makes them ideal for conjugation with peptide drugs. However, the drawbacks of this approach still relate to suboptimal pharmacokinetics and the lack of tissue specificity [91,92].

## 4. Conclusions

Peptide-based drugs are benefiting from the recent advances in science and technology, and we are seeing many novel insights and efforts that accentuate the positive aspects of peptide drugs and aim to erase their inherent weaknesses. This field of investigation is especially thriving from the ongoing breakthroughs in artificial intelligence and deep learning methods, allowing researchers to screen and analyze more potential peptide drugs than was ever possible in the history of drug development. Peptides can also be combined with the PROTAC technology of targeted protein degradation in order to target proteins that cannot be targeted via conventional means. Further research into this area will undoubtably lead to the development of many safe, effective, and selective therapeutics for the pharmaceutical market. Although many excellent reviews focus on the drug design of peptides, ours is the only work that summarizes recent peptide-based PROTACs.

## Figures and Tables

**Figure 1 pharmaceutics-16-01486-f001:**
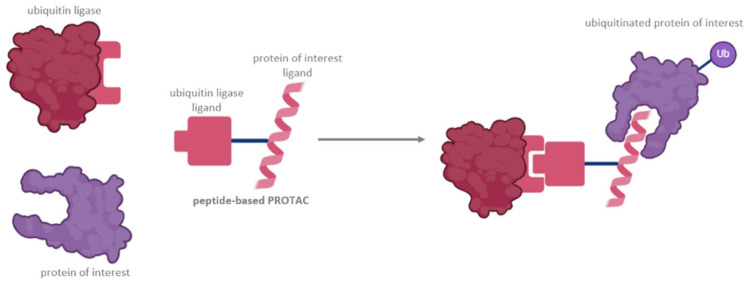
Structure of a peptide-based PROTAC.

**Table 1 pharmaceutics-16-01486-t001:** Peptide-based PROTACs.

Target	Disease	Reference
FOXP3	Cancer	Yang et al. [54]
p300	Prostate cancer	Zhang et al. [55]
Androgen Receptor	Androgenetic alopecia	Ma et al. [56]
CREPT	Pancreatic cancer	Ma et al. [57]
HER2	Breast cancer	Wang et al. [58]
DHHC3	Cervical cancer	Shi et al. [59]
MDM2/MDMX	Colorectal cancer	Chen et al. [60]

**Table 2 pharmaceutics-16-01486-t002:** Pros and cons of peptides as therapeutics.

Pros	Cons
Small size, easy synthesis and modification	Low membrane permeability
High selectivity and specificity	Cellular access challenges
Structural complexity	Low stability and susceptibility to proteolysis
Low immunogenicity and toxicity	Short plasma half-life and rapid renal clearance
Can be conjugated with other molecules for stability	Solubility issues
Potential for site-selective modification	Tissue heterogeneity

## Data Availability

Not applicable.

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
