# Peer review of "Recent Advances in Peptide Drug Discovery: Novel Strategies and Targeted Protein Degradation"

_pharmaceutics, 2024, doi:10.3390/pharmaceutics16111486_

Round 1

Reviewer 1 Report

Comments and Suggestions for Authors

1. The authors are advised to improve the abstract to highlight the unique perspectives of the review.

2. The authors need to use visual aids such as diagrams and flowcharts to improve the manuscript. For example, they may add a schematic diagram of the PROTAC mechanism, comparing traditional and novel methods in peptide drug discovery.

3. The authors need to improve their discussion of challenges in peptide drugs (e.g., bioavailability, and stability).

4. The authors are advised to highlight the limitations and real-world applicability of artificial intelligence in peptide delivery.

5. The reference style needs to be consistent.

6. The manuscript contains several typographical errors and formatting inconsistencies in abbreviation.

Author Response

  1. The authors are advised to improve the abstract to highlight the unique perspectives of the review.

We thank the Reviewer for the suggestion and have changed the abstract to be more in line with the novelty of the text.

  1. The authors need to use visual aids such as diagrams and flowcharts to improve the manuscript. For example, they may add a schematic diagram of the PROTAC mechanism, comparing traditional and novel methods in peptide drug discovery.

We agree with the Reviewer and have added a Figure with the structure of a peptide-based PROTAC molecule.

  1. The authors need to improve their discussion of challenges in peptide drugs (e.g., bioavailability, and stability).

We have expanded the discussion with a few sentences on the challenges of peptide-based drugs in the Discussion section of the text as directed.

  1. The authors are advised to highlight the limitations and real-world applicability of artificial intelligence in peptide delivery.

We have added to the Discussion, a passage describing the limitations of using AI for research into peptide drug discovery and delivery as advised by the Reviewer.

  1. The reference style needs to be consistent.

The reference style has been changed to the ACS Chemical Biology style that MDPI uses in its publications. Brackets within the text containing references have also been standardized. We thank the Reviewer for pointing this out.

  1. The manuscript contains several typographical errors and formatting inconsistencies in abbreviation.

We thank the Reviewer for pointing this out and have corrected the errors.

Reviewer 2 Report

Comments and Suggestions for Authors

Authors summarized strategies for peptide drug discovery and some PROTACs. The manuscript is well organized and I suggest its acceptance for publication after some minor revisions.

1. I suggest Authors add some figures to clearly show the examples mentioned in the manuscript, such as the structures of PROTACs of FOXP3, MDM2 and CREPT

2. Authors add more discussion about pros and cons of peptide-based PROTACs targeting different proteins.

Author Response

Authors summarized strategies for peptide drug discovery and some PROTACs. The manuscript is well organized and I suggest its acceptance for publication after some minor revisions.

  1. I suggest Authors add some figures to clearly show the examples mentioned in the manuscript, such as the structures of PROTACs of FOXP3, MDM2 and CREPT

We thank the Reviewer for the suggestion and have included a figure delineating the structure of a peptide-based PROTAC molecule.

  1. Authors add more discussion about pros and cons of peptide-based PROTACs targeting different proteins.

We agree with this comment from the Reviewer and have now added a subsection in the Discussion that addresses the benefits, limitations, and impact of peptide-based PROTACs in more detail.

Reviewer 3 Report

Comments and Suggestions for Authors

The authors present a good review about the up-to-date strategies and advancements in peptide drug discovery. They covered important and novel technologies including multi-omics, artificial intelligence, and PROTAC. The review is well written and well organized.

This work merits publication after minor corrections.

1.       The authors need to mention the wet lab methods in the abstract as a section that is also covered in this review.

2.       In conclusion, authors need to emphasize on the potential impact of PROTAC in peptide drug discovery since it is the main theme of the review.

3.       Although, the research covers an important and hot area of drug discovery, it is presented more in the form of mini review. I would suggest the following to optimize the impact and the scientific soundness of the review:

·         Adding a section about recent advances in therapeutic delivery of peptide-based drugs, examples, challenges and reflection.

·         Increasing the number of presented research in each section.

·         Increasing the number of references.

Author Response

The authors present a good review about the up-to-date strategies and advancements in peptide drug discovery. They covered important and novel technologies including multi-omics, artificial intelligence, and PROTAC. The review is well written and well organized.

This work merits publication after minor corrections.

  1. The authors need to mention the wet lab methods in the abstract as a section that is also covered in this review.

We have modified the Abstract to reflect the Reviewer’s valuable suggestion.

  1. In conclusion, authors need to emphasize on the potential impact of PROTAC in peptide drug discovery since it is the main theme of the review.

We thank the Reviewer for this suggestion and have now modified the Conclusion as directed. We have also added a subsection in the Discussion that tackles the impact of peptide-based PROTACs, as well as their pros and cons.

  1. Although, the research covers an important and hot area of drug discovery, it is presented more in the form of mini review. I would suggest the following to optimize the impact and the scientific soundness of the review:
    Adding a section about recent advances in therapeutic delivery of peptide-based drugs, examples, challenges and reflection.

We agree that the article would benefit from a section delineating some of the novel delivery methods for peptide-based drugs, and have now added a section corresponding to this topic in the main text.

4. Increasing the number of presented research in each section.

We have added several wet-lab and computational methods to the main text, as well as peptide-based PROTACs, to each section of the main text as suggested by the Reviewer.

5. Increasing the number of references.

We have added more references that fit the review, as directed by the Reviewer.